# A conformation-selective monoclonal antibody against a small molecule-stabilised signalling-deficient form of TNF

Daniel J. Lightwood [1✉], Rebecca J. Munro[1], John Porter [1], David McMillan[1], Bruce Carrington[1], Alison Turner[1], Anthony Scott-Tucker[1], Elizabeth S. Hickford [1], Antje Schmidt[1], David Fox III [2], Alison Maloney[1], Tom Ceska[1], Tim Bourne[1], James O'Connell [1] & Alastair D. G. Lawson[1]

We have recently described the development of a series of small-molecule inhibitors of human tumour necrosis factor (TNF) that stabilise an open, asymmetric, signalling-deficient form of the soluble TNF trimer. Here, we describe the generation, characterisation, and utility of a monoclonal antibody that selectively binds with high affinity to the asymmetric TNF trimer–small molecule complex. The antibody helps to define the molecular dynamics of the apo TNF trimer, reveals the mode of action and specificity of the small molecule inhibitors, acts as a chaperone in solving the human TNF–TNFR1 complex crystal structure, and facilitates the measurement of small molecule target occupancy in complex biological samples. We believe this work defines a role for monoclonal antibodies as tools to facilitate the discovery and development of small-molecule inhibitors of protein–protein interactions.

[1] UCB Pharma, 208 Bath Road, Slough SL1 3WE, UK. [2] UCB Pharma, 7869 NE Day Road W, Bainbridge Island, WA 98110, USA. ✉email: Daniel.Lightwood@ucb.com

Tumour necrosis factor (TNF) is an important therapeutic target for treating a range of autoimmune and inflammatory disorders. The biology of TNF and its role in disease have been extensively studied and reviewed elsewhere[1]. TNF is a pleiotropic cytokine that exerts broad functional activity via binding and activation of two distinct receptors: TNF receptor 1 (TNFR1) and TNF receptor 2 (TNFR2)[2]. TNFR1 is ubiquitously expressed and is activated by both soluble and membrane-associated TNF. Expression of TNFR2 is restricted to specific cell types, such as immune cells, neurons, and endothelial cells and is primarily activated by membrane-associated TNF[3]. It has been proposed that TNFR1 primarily promotes the TNF-induced inflammatory response, whereas TNFR2 controls local homeostatic effects[4]. This study therefore focused on the TNF–TNFR1 interaction.

The development of a number of orally active small molecule inhibitors of human TNF that exhibit activity both in vitro and in vivo has recently been described[5]. The small molecule compounds bind in a pocket at the centre of the TNF trimer and stabilise a conformation where a single TNF monomer subunit is perturbed, thereby inducing a molecular asymmetry in the trimer which in turn affects TNFR1 receptor interaction stoichiometry and subsequent signalling. Data from analytical size exclusion chromatography suggests that the compound-bound asymmetric TNF trimer only associates with two TNFR1 molecules instead of three. This is consistent with the crystal structure of the TNF-small molecule complex which identified significant distortion at one of the three receptor binding interfaces. Rather than inducing a conformational change, molecular dynamic simulations support a mechanism whereby small molecules stabilise a naturally occurring signalling-deficient conformer of TNF.

In a subsequent study with the same small molecule inhibitor series, crystallography and solution-based techniques enabled characterisation of the mechanism of action[6]. Although determination of a compound-bound human TNF-human TNFR1 was not possible, the crystal structure of compound-bound mouse TNF in complex with human TNFR1 revealed a distorted TNF trimer with two copies of receptor bound. Interestingly, the bound receptors existed as dimers, an observation which, along with data from a solution-based network assembly assay, led to the formation of a model for TNF-TNFR1 network signalling and its inhibition by small molecules.

To facilitate further characterisation of these small molecule compounds in a variety of contexts, we here generate a monoclonal antibody, known as CA1974, that selectively binds with high affinity to the TNF trimer–small molecule complex. We solve a crystal structure of the conformation-selective antibody in complex with TNF trimer, small molecule inhibitor, and TNFR1 extracellular domain. The structure confirms other studies suggesting that the small molecule inhibitors stabilise a perturbed conformation of the TNF trimer, resulting in a reduction in receptor stoichiometry, which likely leads to the subsequent inhibition of signalling[5,6]. Furthermore, we show that the antibody, in addition to providing insights into the structure and dynamics of TNF and the mode of action of the small molecules, has potential utility as a tool for measuring target occupancy. This may enable increased understanding of target dynamics and PK-PD effects in preclinical models and when moving into learn-phase clinical studies to confirm proof of mechanism in humans. Collectively, this work highlights the utility of monoclonal antibodies as reagents to support discovery and clinical development of small molecule drug inhibitors of protein–protein interactions.

## Results

**Antibody discovery and specificity determination.** We employed a high-throughput single B-cell platform[7] to efficiently

screen the antibody repertoire from five rats which had been immunised with human TNF pre-complexed with the small molecule inhibitor UCB-9260. A monoclonal antibody, CA1974, was identified and shown to have excellent selectivity for the TNF–small molecule complex compared to the apo form of TNF. Consistent with previous observations that small molecules do not ablate engagement with all three receptor subunits[5,6], binding specificity was initially demonstrated using a human TNFR1-capture ELISA in order to present the distorted A/C interface of the trimer. As shown in Fig. 1a, recombinant Fab fragment of CA1974 bound specifically to TNF complexed with either of the small molecules UCB-9260 or UCB-8733.

To determine if the small molecule inhibitor–TNF complex was able to bind to TNFR1 on the cell membrane, we performed flow-cytometry experiments using human embryonic kidney (HEK)-293 Jump In cells which overexpress TNFR1 after induction with doxycycline (Supplementary Fig. 1) and CA1974

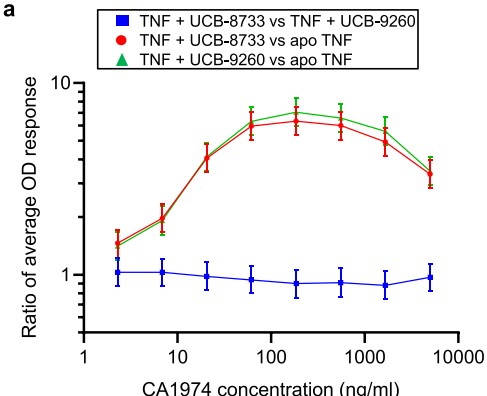

**a**

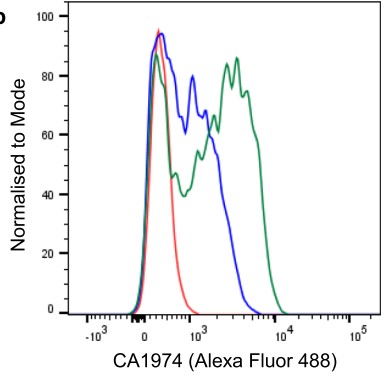

**b**

**Fig. 1 CA1974 binding to human TNF in complex with small molecule inhibitors. a** TNF–small molecule (**UCB-8733** and **UCB-9260**) complex or apo TNF (blue) were captured via TNFR1 which was pre-coated onto a 384-well ELISA plate. CA1974 Fab at a range of concentrations was then added to the plate to detect TNF–small molecule complex. Comparisons are presented as ratios of geometric means (points) and 95% confidence intervals for the ratios (error bars), i.e., Comparison of TNF-**UCB-8733** complex and TNF-**UCB-9260** (blue); Comparison of TNF-**UCB-8733** complex and apo TNF (red); Comparison of TNF-**UCB-9260** complex and apo TNF (green). **b** Human TNF-**UCB-9260** complex at 25 ng/ml (blue), 250 ng/ml (green) or apo TNF (250 ng/ml) with DMSO (0.4%) (red) was added to HEK-293 Jump In cells expressing human TNFR1. CA1974 IgG at 10 μg/ml was used to detect TNF–small molecule complex bound to the surface of cells using flow cytometry. The binding was revealed using a goat anti-mouse IgG Alexa Fluor 488 secondary antibody. Data is shown for cells present in a gate which defines viable single cells and excludes doublets and other cell debris (Supplementary Fig. 1). Cell count (y-axis) is normalised to mode. Source data are provided as a Source Data file.

**Table 1 CA1974 Surface plasmon resonance binding kinetics.**

| TNF/compound type | $k_a$ (M$^{-1}$ s$^{-1}$) ($n=2$) | | $k_d$ (s$^{-1}$) ($n=2$) | | $K_D$ (M) ($n=2$) | | Fold difference $K_D$ |
|---|---|---|---|---|---|---|---|
| Apo human TNF | 1.15E + 05 | 1.12E + 05 | 7.55E-04 | 5.82E-04 | 6.58E-09 | 5.18E-09 | |
| Human TNF + **UCB-8733** | 2.70E + 05 | 2.75E + 05 | <1E-5* | <1E-5* | 3.71E-11 | 3.64E-11 | 159 |
| Human TNF + **UCB-9260** | 2.15E + 05 | 2.18E + 05 | 2.10E-05 | 2.46E-05 | 9.75E-11 | 1.13E-10 | 56 |
| Apo mouse TNF | 1.29E + 05 | 1.28E + 05 | 6.28E-03 | 6.82E-03 | 4.88E-08 | 5.34E-08 | |
| Mouse TNF + **UCB-8733** | 3.38E + 04 | 3.71E + 04 | <1E-5* | 2.67E-05 | 2.96E-10 | 7.20E-10 | 111 |
| Mouse TNF + **UCB-9260** | 3.69E + 04 | 3.91E + 04 | 1.33E-04 | 1.46E-04 | 3.60E-09 | 3.74E-09 | 14 |
| Apo cyno TNF | 1.09E + 05 | 1.15E + 05 | 1.23E-03 | 1.08E-03 | 1.14E-08 | 9.38E-09 | |
| Cyno TNF + **UCB-8733** | 1.32E + 05 | 1.33E + 05 | <1E-5* | <1E-5* | 7.55E-11 | 7.52E-11 | 137 |
| Cyno TNF + **UCB-9260** | 1.14E + 05 | 1.14E + 05 | <1E-5* | <1E-5* | 8.76E-11 | 8.76E-11 | 118 |

On-rate (ka), off-rate (kd), and affinity constant (K$_D$) are shown for $n=2$ independent experiments with fold difference between K$_D$ values in the presence and absence (apo TNF) of compound being calculated from Geometric mean K$_D$ values.
*kd of 1E-05 s$^{-1}$ is at the limit of detection for the device under the conditions outlined in the method. Cyno is cynomolgus monkey.

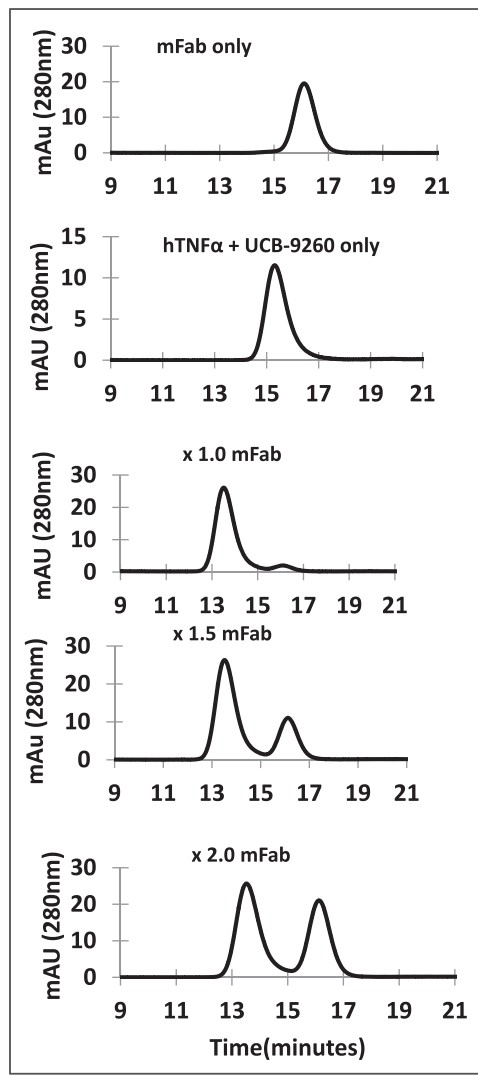

**Fig. 2 Stoichiometry assessment of the CA1974-TNF–small molecule complex.** A Fab fragment of CA1974 was incubated at various molar-ratios with TNF-**UCB-9260** small molecule complex and then analysed by size-exclusion-HPLC. The appearance of only a single high-molecular-weight peak (at approx. 13.5 min) suggests the stoichiometry was 1 Fab: 1 TNF trimer. Source data are provided as a Source Data file.

IgG as a detection agent. We demonstrated that compound-bound TNF can be detected in complex with receptor on the surface of a cell (Fig. 1b). No cell binding was observed following addition of apo TNF in the absence of compound.

Furthermore, to accurately measure the level of specificity of CA1974, we performed affinity measurements by Biacore using human, cynomolgus monkey, and mouse TNF with and without the small molecule inhibitors **UCB-9260** and **UCB-8733**. As shown in Table 1, human and cynomolgus TNF–small molecule complexes exhibited over a 100-fold improved K$_D$ compared to apo TNF (except in the case of human TNF-**UCB-9260** which showed a 56-fold enhanced K$_D$). For those complexes which exhibited 100-fold enhanced K$_D$ versus apo TNF, the measured off-rates (kd) were close to the accurate limit of detection of the Biacore device (1.00E-05 s$^{-1}$), so the difference in affinity between the TNF–small molecule complex and apo TNF may have been even greater than the 2-logs recorded. Over 10-fold lower affinity was observed for antibody binding to mouse TNF–small molecule complex compared to human and cynomolgus complexes. However, an approximate 100-fold greater K$_D$ for mouse TNF–**UCB-8733** complex compared to the mouse apo TNF was observed.

**Stoichiometry assessment by size exclusion (SE)-HPLC.** We wanted to understand the nature of the interaction between the antibody and the perturbed TNF trimer. Structures of TNF in association with small-molecule indicate that the normally symmetric trimer becomes distorted, resulting in a breakdown in trimerous symmetry. A prominent feature of this perturbation is the generation of an expanded cleft at the altered interface between two of the monomer subunits (subunits A and C)[5,6]. We rationalised that this probably represented a region of the trimeric molecule which could be bound by the monoclonal antibody. We tested this theory by performing SE-HPLC to assess complex formation prior to crystallography (see below) and to determine stoichiometry of binding.

Fab fragment of CA1974 was incubated at various molar-ratios with TNF-**UCB-9260** small molecule complex and then analysed by SE-HPLC. As can be seen in Fig. 2, at equimolar concentrations of Fab and TNF-small molecule complex, a new single, higher molecular weight, peak corresponding to Fab bound to TNF trimer-compound complex was evident (retention time of approx. 13.5 min) (representing 90.3% total protein). At 1.5x and 2x molar excesses of Fab, there was no change to the size of the complex (as judged by retention time) and there was a respective increase in the area of the peak present at approximately 15.5 min, probably representing free Fab. Based on these observations, it seems highly likely that the stoichiometry was 1 Fab: 1 TNF trimer–small molecule complex. This observation was consistent with the theory that a single epitope on the TNF trimer, which arises or is stabilised through small molecule binding, is recognised by the antibody.

**Structure of the antibody-TNF–small molecule-TNFR1 complex.** In order to confirm that the CA1974 antibody bound a single epitope at the distorted TNF interface and to study the

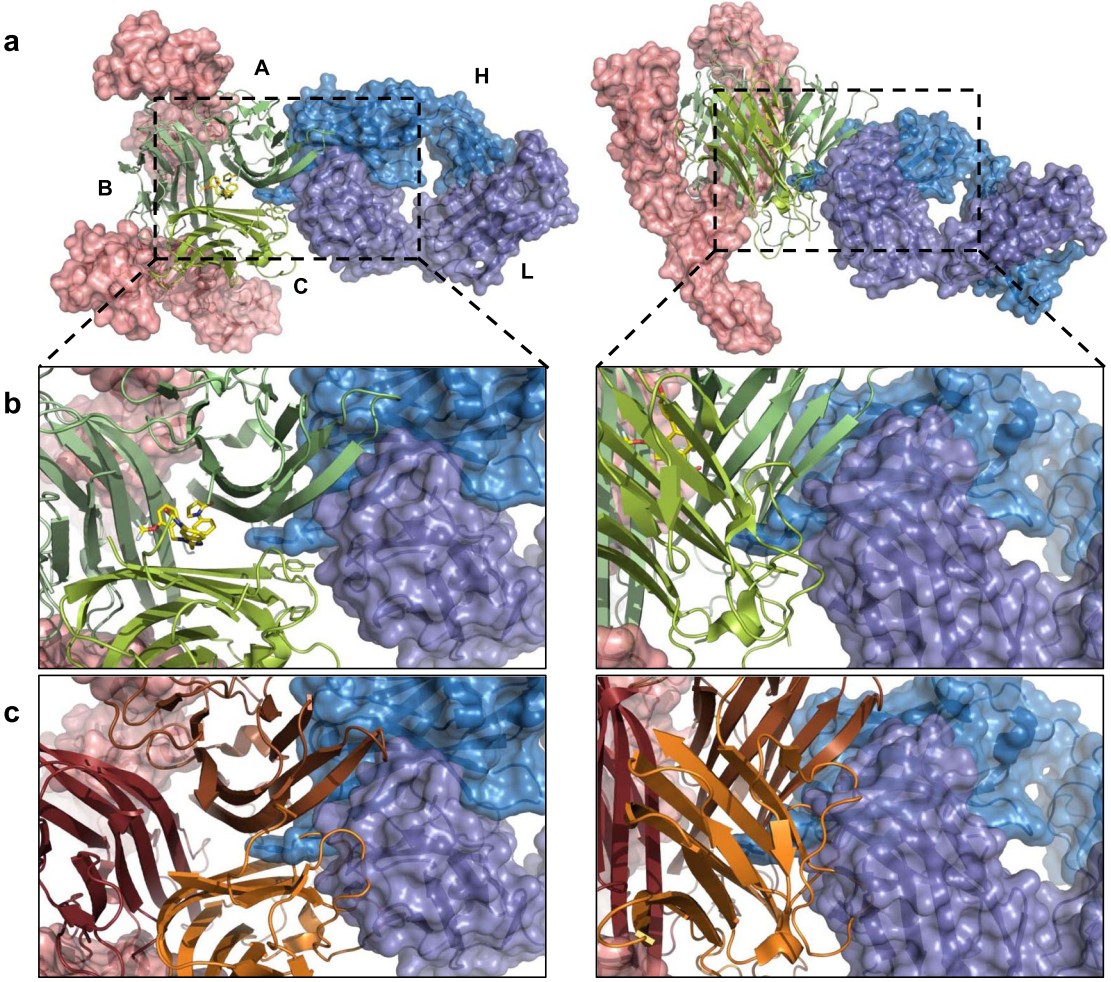

**Fig. 3 Structure of CA1974 Fab-human TNF-UCB-8733-TNFR1 complex. a** Trimeric human TNF (green ribbons, monomers assigned A, B, & C) with **UCB-8733** bound (yellow sticks), in complex with CA1974 Fab (blue surface rendered) and two copies of human TNFR1 (pink surface rendered), viewed from the top and side (left and right images respectively). **b** Detailed view of the CA1974 Fab-human TNF (with **UCB-8733** bound) interface viewed from the top and side (left and right images respectively). **c** Detailed view of a structural model where compound-bound human TNF has been replaced by apo human TNF (brown ribbons) viewed from the top and side (left and right images respectively). This highlights side-chain clashes between the CA1974 Fab and symmetric apo TNF.

effect of small molecule on receptor stoichiometry, we generated a complex consisting of CA1974 antibody Fab fragment, human TNF trimer, **UCB-8733** small molecule inhibitor, and human TNFR1 extracellular domain (Supplementary Fig. 2). The structure was then solved by X-ray crystallography (PDB code 7KPB).

All components of the complex were well resolved at 3.0 Å, clearly indicating a single copy of CA1974 bound at the distorted interface of the TNF trimer and copies of hTNFR1 bound at the non-distorted receptor-binding sites (Fig. 3a, Supplementary Fig. 3).

The main contact between TNF and the Fab is via the heavy chain and the displaced TNF monomer A. Heavy chain CDRs 2 and 3 form a contact with the outer face of monomer A and CDR3 protrudes deeply into the open cleft of the distorted A/C receptor-binding site (Fig. 3b). The positioning of CDR3 in this cleft results in the only contact between the CA1974 heavy chain and TNF monomer C where pi stacking occurs between Tyr103 of CDR3 and Tyr115 of monomer C.

The light chain of CA1974 also forms a substantial contact with TNF, bridging the distorted A/C receptor-binding cleft. The light chain CDR2 contacts with monomer C exclusively, towards the bottom of the cleft (thin end of TNF). Only two residues of CDR1 contact TNF, one binding monomer C and the other

monomer A, on the inner faces of the cleft. Light chain CDR3 associates solely with monomer A, interacting with residues towards the top of the cleft (thick end of TNF) (Fig. 3b).

We have demonstrated that CA1974 is capable of binding apo TNF, so it was of interest to model the symmetric apo TNF structure (PDB code 1TNF) bound with CA1974. When symmetric apo TNF was aligned with TNF in our complex structure (alignment done through monomer A as this forms the main contact with CA1974) there were clear clashes between apo TNF and the Fab. Unsurprisingly, given the degree of distortion in the compound-bound TNF, there were a number of very significant structural clashes between several surface loops on monomer C of apo TNF and the CDR and framework regions of CA1974 (Fig. 3c). When aligning through monomer A, as done here, the relative repositioning of monomer C can be exemplified by the 7.3 Å shift in position of Tyr115 preventing the pi stacking described above from occurring (Supplementary Fig. 4). These data suggest that CA1974 is unable to bind to symmetric apo TNF.

The data here reveal the structure of human TNF with its cognate human receptor. As anticipated, the nature of the ligand–receptor interaction is very similar to those observed previously for closely related TNF/TNFR combinations (PDB

codes: 1TNR, 3ALQ & 7KP8)[6,8,9]. In particular, an overlay of the TNF and receptors from this structure with those from the compound-bound mouse TNF/human TNFR1 structure (7KP8)[6] reveal a striking similarity: RMSD = 0.75 Å (Supplementary Fig. 5). The remarkable similarity in the two structures strengthens the evidence presented previously that the compounds block TNF signalling by stabilising a distorted TNF trimer that binds receptor with a reduced stoichiometry of two receptors per trimer[5,6].

To better understand any potential influence of CA1974 binding on the conformation of TNF, compound-bound TNF from the CA1974 plus receptor complex structure was compared to the structure of TNF plus **UCB-8733** in the absence of CA1974 and receptors (PDB code 7KPA) (Fig. 4, Supplementary Fig. 6). Overall the homology was very high, with an RMSD = 0.43 Å with no significant variation in the degree of distortion at the A/C receptor binding interface (Supplementary Table 1). Variation was seen in loops containing Tyr87 (Fig. 4a red circles) but this is perhaps not unexpected, given that the loops are flexible and poorly ordered in the compound-only structure[5]. Whereas in the structure with Fab and receptors the loops are stabilised through protein contacts with either receptors or Fab (Fig. 4b). This suggests that the CA1974 antibody recognises the true small molecule-stabilised form of the TNF trimer and does not induce significant further structural rearrangements as a result of binding.

As described, CA1974 also recognised cynomolgus monkey and, to a lesser degree, mouse TNF–small molecule complexes.

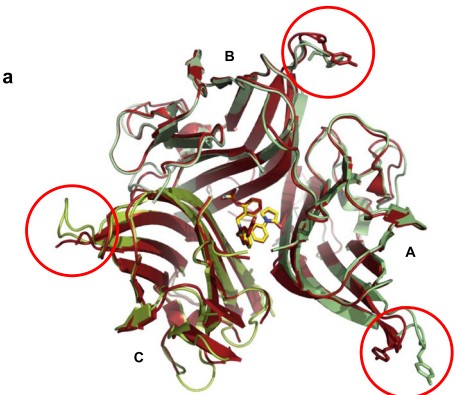

RMSD = 0.43Å

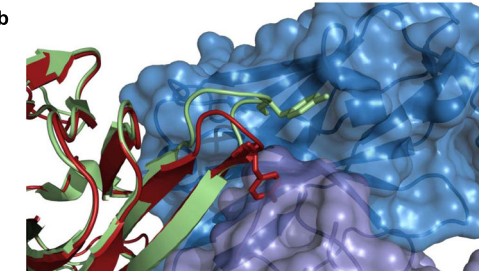

**Fig. 4 Comparison of human TNF-UCB-8733 free and antibody-complexed structures. a** Alignment of human TNF from the compound (**UCB-8733**)-only structure (red ribbons) with human TNF (plus **UCB-8733**) from the CA1974 Fab-human TNF-**UCB-8733**-TNFR1 complex structure (green ribbons) (Fab and receptors have been removed for clarity). Monomers A, B, and C are labelled. Red circles highlight a region of flexibility containing Tyr87 (sticks) on each monomer. **b** Detail showing the change in position of Tyr87 (sticks) in monomer A when Fab CA1974 is bound (green ribbon) compared to its position in the human TNF **UCB-8733**-only structure (red ribbon). Further details of residue movements are shown in Supplementary Table 2.

To understand the nature of the interaction with TNF from these different species, we determined the epitope on the human TNF trimer and identified those residues that were different in the cynomolgus and mouse molecules (Fig. 5; Supplementary Table 2). As expected, the mouse TNF trimer possessed five residues which differed to the human molecule within the epitope: I83F, T89E, I97V, I136V, and R138L. Interestingly, all of these mapped onto subunit A of the trimer. Only one change was observed between human and cynomolgus monkey TNF trimer (R138L), and this same change was present in the mouse. Given the comparable affinity of CA1974 to human and cynomolgus monkey TNF–small molecule complexes, it is likely that I83F, T89E, I97V and I136V are sufficient to confer reduced affinity towards the mouse TNF–small molecule complex.

**Assessment of CA1974 as a target occupancy reagent**. One of the main objectives of this work was to generate a reagent that would allow for measurement of target occupancy in plasma samples taken from animals and eventually humans. An ELISA was established utilising CA1974 as a capture reagent to measure a range of concentrations of recombinant human TNF-**UCB-8733** complex diluted in neat human plasma. As can be seen in Fig. 6, TNF–small molecule complex could be detected at 25 pg/ml and above in neat plasma (with probability of at least 0.999) and there was no background cross-reactivity observed with apo TNF. This demonstrates that CA1974 may be a useful reagent for the measurement of target occupancy in complex biological samples.

**Binding of CA1974 to apo TNF**. Despite the structural analyses suggesting that CA1974 is unlikely to interact with the symmetric form of apo TNF due to major side-chain clashes that would likely prevent association, Biacore measurements (Table 1) demonstrated that there was a relatively low-affinity interaction of CA1974 with the apo TNF molecule. The interaction of antibody with apo TNF exhibited both slower on-rate and faster off-rate kinetics compared to the interaction with the stabilised TNF–small molecule complex. Interestingly, as observed with the TNF trimer–small molecule complex, incubation of Fab with apo TNF also resulted in the generation of a higher molecular weight species predicted to be a 1:1 complex (21% of total protein) in addition to free unbound Fab and TNF (Supplementary Fig. 7). These data suggest that the antibody may recognise a similarly perturbed open form of the trimer that exists naturally but is only represented by a small fraction of the TNF population and is potentially only sampled transiently. This supports the conclusions made in previous studies using molecular dynamic simulations[5] and DEER analysis of spin-labelled trimer[10].

## Discussion

We have recently described the generation of small molecule inhibitors of TNF[5,6]. These inhibitors bind at the centre of the TNF trimer and stabilise a perturbed form of the TNF molecule that is unable to bind one of the three receptor dimers that are normally engaged and, as a result, has reduced signalling capacity. To facilitate an investigation into the mode of action of these small molecules in a variety of assays and to allow for target occupancy measurements to be made in biological samples, we generated a conformation-selective monoclonal antibody, known as CA1974, which bound with high affinity to the distorted TNF–small molecule complex.

Initial specificity screens were based on detection of TNFR1-captured TNF–small molecule complex in an ELISA with protein (Fig. 1a). We utilised this assay to avoid the potential deleterious effects of biotinylation or direct immobilisation of TNF on plastic,

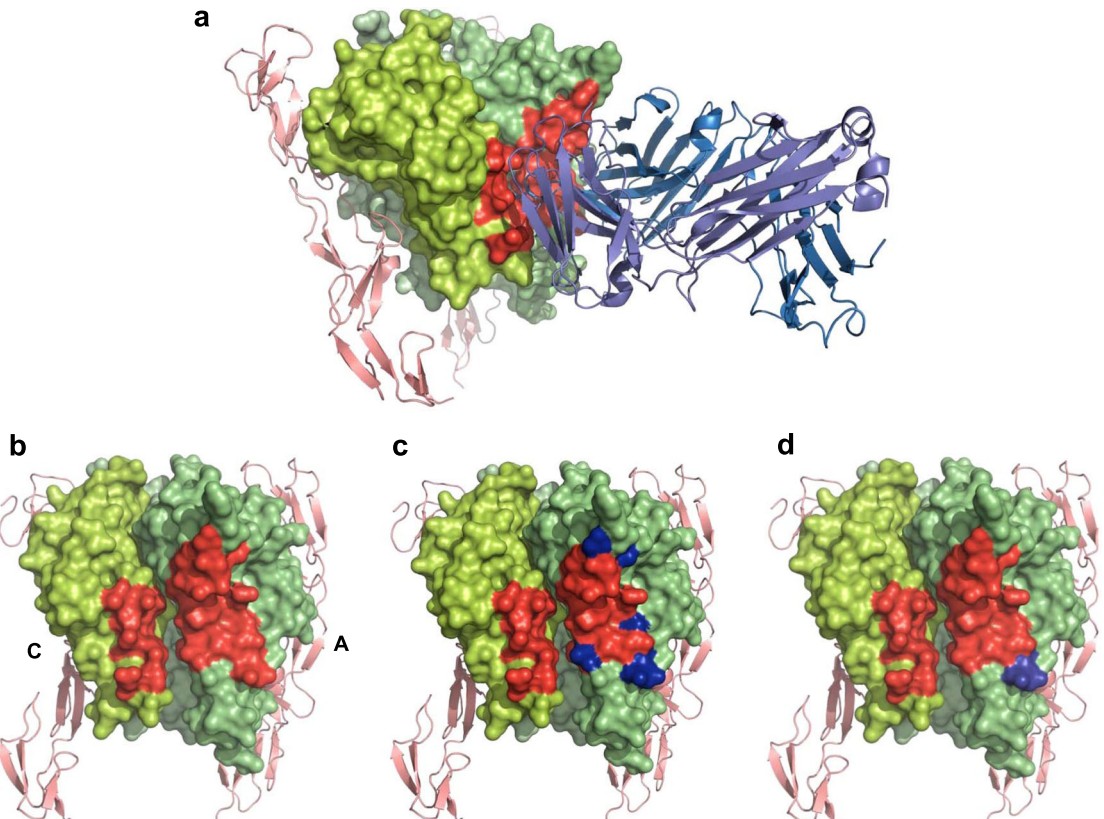

**Fig. 5 Detail of the epitope of CA1974 Fab. a** Complex of human TNF (green surface rendered), CA1974 Fab (blue ribbons), and human TNFR1 (pink ribbons) with epitope residues highlighted in red. **b** Head-on view of the CA1974 Fab epitope spanning monomers A & C on human TNF (highlighted in red). For clarity CA1974 Fab has been removed and monomers A and C have been labelled. **c** CA1974 Fab epitope with residues that vary in mouse TNF marked in blue. **d** Equivalent view with residues that vary in cynomolgus monkey TNF marked in blue. Residues that vary between the species are highlighted in Supplementary Table 2.

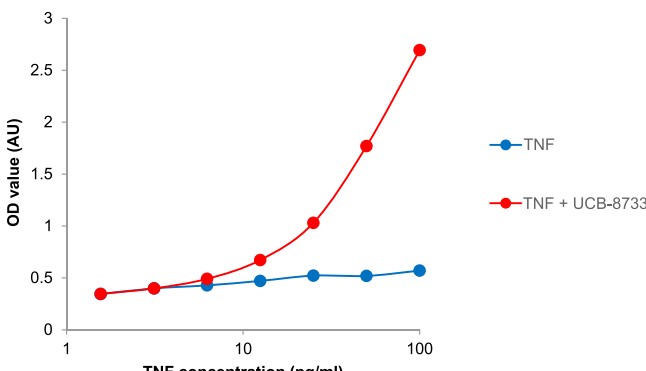

**Fig. 6 CA1974 represents a suitable reagent for measuring target occupancy.** CA1974 was used as a capture reagent to measure TNF complexed with **UCB-8733** small-molecule inhibitor using a high-sensitivity ELISA, with a commercial anti-TNF antibody as the detection reagent. TNF pre-incubated with either excess compound or DMSO was spiked into neat human plasma (depleted of endogenous TNF) and diluted to various concentrations and used in the assay. TNF–small molecule complex signal (red) was significantly higher than apo–TNF background (blue), with probability of at least 0.999 at TNF concentrations of 25 pg/ml and above. Data from representative experiment. Source data from this and other experiments showing utility of CA1974 in measuring target occupancy are provided as a Source Data file.

and we already had data suggesting that TNF–small molecule complex could bind the TNFR1 extracellular domain molecule[5,6]. We then went on to show that the small molecule inhibitor-TNF complex was able to bind to TNFR1 on cells via flow cytometry using CA1974 IgG as a detection agent. Confirmation of binding in both assays supported observations made by McMillan et al.[6], where it was hypothesised that the small molecule inhibitor-perturbed TNF was able to bind to receptor on cells but that signalling was defective through disruption of the normal cross-linking architecture that is thought to arise on binding of symmetric TNF trimer to receptor on cells[11–14].

Further evidence for conformational selectivity was gained through the use of Biacore to determine binding kinetics of CA1974 against apo and small molecule-complexed TNF in solution (Table 1). We showed examples of human, cynomolgus monkey, and mouse TNF–small molecule complexes exhibiting at least a 100-fold improved $K_D$ compared to apo TNF. Both the on-rate and off-rate kinetics were improved in the presence of the small molecule. Having cross-reactivity to preclinical species such as mouse and cynomolgus monkey is attractive and highlights the utility of this antibody reagent. CA1974 allows the parallel in vitro characterisation of the drug in multiple species and facilitates the translation of data generated in animal studies to human clinical trials.

In particular, CA1974 enables the measurement of target occupancy in biological samples taken from animals and eventually humans. We were able to measure human TNF–small molecule complex at concentrations as low as 25 pg/ml in human

plasma without measurable reactivity with apo TNF (Fig. 6). TNF levels in serum and synovial fluid of rheumatoid arthritis patients vary considerably. However, TNF concentrations above 30 pg/ml have been frequently reported[15–17] and synovial fluid concentrations of TNF in the ng/ml range have also been published[18,19]. This suggests that the assay described here is likely to be able to detect clinically relevant concentrations of TNF in complex with small molecule inhibitor within natural biological matrices.

To understand the mechanism with which the antibody bound with such remarkable selectivity, we first performed SE-HPLC experiments to determine the stoichiometry of antibody (Fab fragment) binding to TNF trimer–small molecule complex. The data clearly suggested that the TNF trimer–small molecule complex accommodates a single Fab fragment. This observation was consistent with the hypothesis that a single epitope on the TNF trimer, which arises or is stabilised through small molecule binding, is recognised by the antibody reagent.

In order to understand the mechanism with which CA1974 antibody recognised the distorted TNF trimer and to gain further insights into the interaction of the perturbed trimer with receptor, we generated an X-ray crystal structure of a complex consisting of CA1974 Fab fragment, human TNF trimer, **UCB-8733** small molecule inhibitor, and human TNFR1 extracellular domain. Previous attempts to obtain a structure of the human TNF–TNFR1 complex had failed[6]. The elucidation of a crystal structure was only made possible through the inclusion of the small molecule inhibitor and the conformation-selective antibody CA1974.

Consistent with previous work suggesting that small molecules reduce TNF:TNFR1 stoichiometry from 1:3 to 1:2, which leads to a disruption in productive receptor assembly and signalling, the work described here, using a fully human TNF-receptor system, supports that conclusion. As predicted, the structure revealed that CA1974 recognises a unique epitope at the perturbed A/C receptor-binding interface, formed as compound captures TNF in a distorted conformation where monomer A has tilted and twisted down and out.

Interestingly, despite structural models revealing major side-chain clashes with the symmetric form of apo TNF (Fig. 3c), both Biacore and SE-HPLC experiments suggested that CA1974 is able to interact with apo TNF. The affinity of the interaction was typically >100-fold lower than that observed with the TNF–small molecule complex. SE-HPLC suggested that CA1974 interacts with a small fraction (21%) of apo TNF and produces a complex that has elution characteristics consistent with a 1:1 (Fab: apo TNF trimer) complex. Taken together, these observations support the hypothesis that apo TNF naturally adopts an open conformation that allows antibody recognition rather than the antibody binding to a partial epitope on the symmetric molecule. The lower amount of complex that is present in the SE-HPLC experiment compared to the Fab-TNF–small molecule complex and the lower observed affinity for apo TNF could imply that this open conformation is sampled naturally but exists only transiently and that this form of the apo TNF is much less energetically stable than the small molecule-stabilised form and hence the antibody interaction is compromised. These findings support the conclusions made in previous molecular dynamic simulation studies[5] and DEER experiments with spin-labelled TNF which predict 6% of trimers at any one time adopt a naturally sampled asymmetric open state[10]. The presence of CA1974 in the SEC HPLC experiment probably draws the equilibrium of conformational sampling towards the open state, increasing the percentage of asymmetric trimer. The low fraction of open conformer in the apo TNF sample and the reduced affinity compared to the small

molecule-bound form are also likely to account for the lack of signal with apo TNF seen in the ELISA screens (Figs. 1a and 6).

This work highlights the dramatic conformational dynamism of the TNF trimer molecule and raises questions as to the biological significance of this breathing. Such flexibility may be a precursor to trimer dissociation which could be a mechanism with which TNF activity and half-life is regulated in vivo[20,21]. In addition, this open conformation may facilitate monomer exchange[22,23] or be a critical conformational state allowing TNF to efficiently engage with its receptor system in either an activatory or a non-signalling, possibly regulatory, way. Further research is required to establish the role that TNF breathing plays in the function of this highly active and therapeutically relevant cytokine. This work also highlights how antibodies, which define rarely sampled conformations of target proteins, may facilitate small molecule fragment screening[5] by enabling the binding of otherwise entropically disadvantaged compounds (Supplementary Fig. 8) or act as a sensor to detect new chemical matter able to stabilise predefined inactive conformations. Conversely, small molecules, which stabilise rarely sampled conformations of target proteins, may augment the generation of antibodies recognising induced neo-epitopes.

The utility of antibodies that recognise conformational variants of molecules has been highlighted previously[24–27]. However, in the current study we exemplify the use of an antibody as an agent to detect small molecule drug binding to a therapeutic target molecule. The use of immunisation and deep screening of B cells resulted in the generation of a high-affinity antibody molecule that has shown utility in supporting multiple aspects of small molecule discovery for inhibition of protein–protein interactions, a challenging and exciting area of drug research. The antibody has helped define the mode of action and specificity of a novel class of small molecule inhibitors of TNF, facilitated the measuring of small molecule target occupancy, revealed the molecular dynamics of the apo TNF trimer, and acted as a chaperone to enable crystallisation of otherwise recalcitrant systems. In summary, this work highlights the power of monoclonal antibodies, not only as therapeutic entities in their own right, but how they can be used to facilitate the discovery and development of the next wave of small-molecule drugs for treating patients suffering with severe diseases.

## Methods

**Availability of materials**. Most reagents are commercially available and details are provided in the manuscript. Access to proprietary UCB materials may be available on request under an appropriate material transfer agreement.

**Small molecule inhibitors**. **UCB-9260** small molecule inhibitor was generated at UCB with details described in O'Connell et al.[5]. **UCB-8733** small molecule inhibitor was generated at UCB Pharma. Details for this are provided in Supplementary Note 1. A summary of the structure and binding kinetics of **UCB-9260** and **UCB-8733** is given in Supplementary Fig. 9.

**Plasmid construction**. Human TNF (UniProt P01375 residues 77–233), cynomolgus monkey TNF (Uniprot P79337; residues 77–233), and mouse TNF (Uniprot P06804; residues 82–235) genes were codon-engineered in silico using GeneComposer™ for *E. coli* expression, and optimised to balance GC content, and exclude cryptic Shine Dalgarno sequences as well as BamHI and HindIII restriction sites. The final gene inserts were flanked with 5′ GGATCC (BamHI) and 3′ TGATAAGCTT (HindIII is underlined), such that two stop codons follow the C-terminal residue. The final gene inserts were then synthesized by DNA 2.0 and delivered in a shuttle vector. Following synthesis, the gene inserts were digested with BamHI and HindIII and subcloned to vector pEMB54, which is an ampicillin-resistant, arabinose-inducible vector with pMB1 origin of replication and 6XHis-Smt3 under the PBAD promoter, followed by a multiple cloning site that includes BamHI followed by HindIII. After BamHI/HindIII cloning into pEMB54, gene inserts are fused in-frame with the 6XHis-Smt3 sequence. Following digestion of both pEMB54 and the respective gene inserts with BamHI/HindIII, DNA fragments were gel-purified, and ligated into the vector and then the ligation was transformed to chemically competent TOP10 *E. coli* cells (Thermo fisher, catalogue

number C404010). One transformant was mini-prepped per construct and submitted for DNA sequencing of the Open Reading Frame.

**Protein expression and purification.** Briefly, the target specific vector was transformed into TOP10 *E. coli* cells (Thermo fisher, catalogue number: C404010). A starter culture containing 100 µg/ml (final concentration) ampicillin (Teknova) was inoculated with a single colony and grown for 16 h at 37 °C. This was then transferred to 8 litres of terrific broth (Teknova) containing 100 µg/ml (final concentration) ampicillin and grown to OD600 = 0.6. Protein expression was induced by adding arabinose to a final concentration of 0.1% (VWR) and grown for 16 h at 25 °C. The cells were harvested by centrifugation (Beckman) at $6200 \times g$ for 15 min and the pellets were collected and stored at −80 °C.

Cells were resuspended 1 g:4 ml in 25 mM Tris(hydroxymethyl)aminomethane hydrochloride (Tris-HCl) pH 8.0 (Teknova), 200 mM NaCl (Teknova), 0.02% 3-[(3-Cholamidopropyl)dimethylammonio]-1-propanesulfonate (CHAPS) (JT Baker), 50 mM L-arginine (Sigma), 500U of benzonase (Novagen), 100 mg lysozyme (Sigma) and one complete EDTA-free protease inhibitor tablet (Roche). The cells were lysed via sonication (Misonix) and clarified via centrifugation at $142,000 \times g$ for 30 min at 4 °C (Beckman) and filtered with a 0.2 µm bottle-top filter (Nalgene). The supernatant was applied to two 5 ml Ni2+ charged HiTrap Chelating HP (GE Healthcare) columns and the protein eluted with a 500 mM imidazole gradient over 20 column volumes. The fractions of interest were pooled and the His-Smt tag was removed via cleavage with Ubiquitin-like-specific protease 1 (Ulp-1) while dialysing against 2 L of 25 mM Tris pH 8.0 and 200 mM NaCl overnight at 4 °C using 3.5 kDa MWCO snakeskin dialysis tubing. The affinity tag was removed by applying the digested pool over two 5 ml Ni2+ charged HiTrap Chelating HP columns. The flow-through contained the cleaved protein of interest. The protein was concentrated for size exclusion chromatography via centrifugal concentration (Vivaspin Polyethylsulfone, 10 kDa MWCO, Sartorius) to 13.5 mg/ml for injection over a HiPrep 16/60 Sephacryl S-100 HR (GE Healthcare) in 10 mM HEPES (2-[4-(2-hydroxyethyl)piperazin-1-yl]ethanesulfonic acid), pH 7.5 and 150 mM NaCl. Fractions of interest were pooled and concentrated via centrifugal concentration (Vivaspin Polyethylsulfone, 10 kDa MWCO, Sartorius) to 20 mg/ml, aliquoted, and stored at −80 °C.

**Antibody generation.** Five female Sprague Dawley rats (150–180 g) (approx. 8-weeks of age) were immunised subcutaneously with three shots of 50 µg human TNF pre-complexed for 2 h at room temperature (RT) with 50-fold molar excess of **UCB-9260**. Knowledge that the small molecule–TNF interaction had a sufficiently slow off-rate ($k_{off}$ 0.019 s⁻¹) and evidence that the complex could exist in vivo following mouse efficacy models[5] supported the immunisation approach taken here. Complete Freund's adjuvant was used for the first shot, while subsequent immunisations used incomplete Freund's adjuvant in a 1:1 v/v ratio with TNF–small molecule complex. Approval for use of animals for immunisation was provided through the UCB Pharma, UK Animal Welfare and Ethical Review Body (AWERB), and the license was granted by the UK Home Office. At the end of the study the mice were anaesthetised with isoflurane, terminal bleeds taken, and then sacrificed using a Schedule 1 method in accordance with the Animals Scientific Procedures Act (ASPA).

Antibody discovery was essentially performed using a B cell screening method as previously described[7]. Briefly, splenocyte suspensions from the immunised rats containing immune B cells were cultured in 96-well plates at 1000–5000 cells per well in order to induce clonal expansion and antibody secretion. IgG in culture supernatants were then screened in a homogeneous fluorescence-based bead assay for binding to apo human TNF and human TNF in complex with **UCB-9260** (at a 50-fold molar excess). In the homogeneous assay, human TNF (±small molecule) was captured onto beads using human TNFR1-Fc (R&D Systems, catalogue number 372-RI-050) coated with Superavidin beads (Bangs Laboratories) coated with 2 µg/ml of biotinylated goat F(ab)₂ anti-human Fc reagent (Jackson ImmunoResearch, catalogue number 109-066-098). Selective binding to human TNF in complex with small molecule was confirmed by ELISA using TNF in complex with **UCB-9260** and **UCB-8733** small molecules captured onto plates coated with TNFR1-Fc (see below). Antibodies that demonstrated preferential binding to the TNF-compound complex were taken forward for cloning. The Fluorescent Foci method[28] (US Patent 7993864/ Europe EP1570267B1) utilising beads coated with pre-complexed TNF-**UCB-9260** small molecule inhibitor (50-fold excess) was used to identify and isolate antigen-specific B cells from positive wells. Specific antibody variable-region genes were recovered from single cells by reverse transcription (RT)-PCR using heavy and light chain variable- region-specific primers. The following PCR primers were used: CTGGTTTCCAGGCACCAGGTGTGACATCCAGATGACCCAGTCTCC (VK primary forward); CTTTCATATTCAACCTTGGTCAAC (VK primary reverse); TTGTTGCTCTGGTTTCCAGGCACCAGGTG (VK secondary forward); TGGATACAGTTGGTGCAGCATCCGTACGTTTCAGTTCCAGCTTGG (VK secondary reverse); CAGTAACTACCGGTGTCCATTCTGAGGTGAAGCTGTTG GAATCTGG (VH primary forward); GATAGACHGATGGGGSTGTTG (VH primary reverse); TTCTCTTCTTCCTGTCAGTAACTACCGGTGTCCATTC (VH secondary forward); GATGGGGGTGTTGTTTTAGCACTCGAGACAGTGACC AGAGTGCC (VH secondary reverse). Secondary PCR primers contained restriction sites allowing cloning of the variable region into a mouse IgG1 (VH), mouse Fab (VH), or mouse kappa (VK) mammalian expression vector under the control of a

human cytomegalovirus (HCMV) promoter. Heavy and light chain constructs were co-transfected and expressed using the Freestyle 293 expression system (Thermo Fisher, catalogue number K9000-01) following manufacturer's guidelines and recombinant chimeric IgG1 antibody or Fab (composed of rat variable domains and mouse constant domains) expressed in 30 ml cultures. After expression for 7 days, supernatants were harvested and antibody rescreened for selectivity using the specificity assays described above and also tested by Biacore to determine affinity for apo and small molecule-bound TNF (see below). CA1974 was selected as the lead molecule based on its selective binding and affinity profile. To support further analysis, CA1974 IgG and Fab were purified on an AKTA system (GE healthcare) using affinity chromatography (protein A for IgG; protein G for Fab) followed by SE-HPLC to produce a product >98% pure monomer species.

**TNF–small molecule complex ELISA.** In order to identify conformation-specific antibodies in B-cell culture supernatants, HEK-293 (Thermo Fisher, catalogue number K9000-1) culture supernatants or as purified preparations, a binding ELISA was developed. Nunc maxisorp 384-well ELISA plates (Sigma Aldrich, catalogue number P6366) were coated with 10 µl per well of 1 µg/ml human TNFR1-Fc (R&D Systems, catalogue number 372-RI-050) overnight at 4 °C. Plates were washed and then blocked for 1 h with 1% BSA in PBS. After washing plates again, 10 µl of 100 ng/ml apo TNF or TNF pre-incubated with compound at a 50-fold molar excess in 1% DMSO (final concentration) in PBS for 2 h at room temperature, were added to each well of the plate and incubated for a further 1 h at room temperature. After washing again, mouse IgG or mouse Fab at a range of concentrations diluted in 1% BSA in PBS were added to the plate and incubated for 1 h at room temperature. Antibody binding was detected with either an HRP-conjugated goat anti-mouse IgG Fcγ-specific antibody (Jackson ImmunoResearch, catalogue number 115-036-071) or a goat anti-mouse IgG F(ab)2-specific antibody (Jackson ImmunoResearch, catalogue number 115-036-072) at a 1:5000 dilution in 1% BSA in PBS. Binding was revealed with 3,3′,5,5′-Tetramethylbenzidine (TMB) Substrate and absorbance at 630 nm measured on a synergy 2 microplate reader (BioTek).

The OD responses ($n = 2$ experiments, with duplicate data points within each) were analysed on the logarithmic scale to satisfy the analysis assumptions. The linear model included fixed effects of group, concentration, and associated interactions and allowed for experimental differences. Pre-planned comparisons between the groups at each concentration were performed and results of the unadjusted two-sided comparisons are presented as ratios of geometric means and 95% confidence intervals

**Surface plasmon resonance analysis.** Surface plasmon resonance (SPR) was performed at 25 °C using a Biacore T200 (GE Healthcare). Goat anti-mouse IgG Fcγ-specific antibody (Jackson ImmunoResearch, catalogue number 115-006-071) was immobilised on a CM5 Sensor Chip (GE Healthcare) via amine coupling chemistry to a capture level of approx. 6000 response units. HBS-EP buffer (10 mM HEPES pH 7.4, 0.15 M NaCl, 3 mM EDTA, 0.05% (v/v) surfactant P20 (GE Healthcare)) +1% DMSO was used as the running buffer. A 10 µl injection of CA1974 IgG at 1 µg/ml was used for capture by the immobilised anti-mouse Fcγ reagent to create the TNF-binding surface. Human, cynomolgus monkey or mouse TNF at 50 nM was pre-incubated with 2 µM compound (**UCB-9260** or **UCB-8733**) in HBS-EP + (1% DMSO) for 5 h.

A 3-minute injection of TNF ± test compound was passed over captured CA1974 IgG at a flow rate of 30 µl/min. The surface was regenerated at a flow rate of 10 µl/min by two 60 s injections of 40 mM HCl followed by a single 30 s injection of 5 mM NaOH. Double-referenced, background-subtracted binding curves were analysed using the T200 Evaluation software (version 1.0) following standard procedures. Kinetic parameters were determined from the fitting algorithm.

For single-cycle kinetics profiling of **UCB-6876**[5] or **UCB-9260** to either directly immobilised TNF or CA1974-captured TNF, surface plasmon resonance was performed at 25 °C using a BIAcore T200 (GE Healthcare). Human TNF was tethered onto flow-channel (Fc)2 (~2000 RU), CA1974 IgG (>10,000 response units) tethered onto Fc3,4 and Fc1 blank immobilisation on a CM5 Sensor Chip (GE Healthcare) via amine coupling chemistry. The surface was allowed to stabilise in HBS-P buffer (10 mM HEPES pH 7.4, 0.15 M NaCl, 0.005% (v/v) surfactant P20-GE Healthcare) + 5% DMSO was used as the running buffer. Human TNF was flowed over Fc4 to a capture level of ~2,000 RU. Compound **UCB-6876** was flowed over all four work cells in series (15.625 µM, 31.25 µM, 62.5 µM, 125 µM and 250 µM) at 30 µl/min. Compound **UCB-9260** was flowed over all four flow cells in series (1.875 µM, 3.75 µM, 7.5 µM, 15 µM, and 30 µM) at 100 µl/min. Double-referenced, background-subtracted binding curves were produced using the T200 Evaluation software (version 1.0) following standard procedures.

**Binding to TNF–small molecule complex on cells.** Purified recombinant CA1974 IgG was tested for binding to TNF–small molecule inhibitor complex in a flow-cytometry assay using human embryonic kidney (HEK)-293 Jump In cells (Thermo Fisher, catalogue number A15008), which overexpress TNFR1 after induction with doxycycline at 1 µg/ml for 2.5 h (Supplementary Fig. 1a). HEK-293 Jump In cells were trypsinised and incubated for 2 h in medium to allow recovery of digested TNFR1 levels. Human TNF at a range of concentrations was pre-

incubated with 4.0 μM **UCB-9260** or 0.4% DMSO (equivalent concentration of DMSO) for 1 h at 37 °C. The pre-incubation mix was added to a 3-fold v/v excess of the cells, giving a TNF concentration of either 25 or 250 ng/ml. The sample was incubated for 1 h on ice. Cells were washed, fixed (1.5% PFA), and stained with 10 μg/ml CA1974 antibody for 1 h on ice. After washing again, binding was revealed using a secondary anti-mouse-Alexa fluor-488 antibody (Jackson Immu-noResearch, catalogue number 115-545-071). Samples were run on a BD Bio-sciences Canto II Flow Cytometer with three lasers and a standard 4:2:2 configuration. The Alexa fluor-488 was excited by a 488 nm laser and fluorescence collected through the 530/30 BP filter. Gating of single viable cells (excluding doublets and other cell debris) was made using a forward scatter versus side scatter plot (Supplementary Fig. 1b). TNFR1 expression within the population of gated cells following doxycycline at 1 μg/ml for 2.5 h was assessed in a separate experiment using an anti-human TNFR1 monoclonal antibody at 10 μg/ml (R&D Systems, catalogue number MAB225) followed by staining with goat anti-mouse-Alexa488 secondary antibody (Molecular Probes, catalogue number A-11001) at 1:200 dilution. This showed heterogeneous expression of TNFR1 as well as the presence of a population which appeared negative for receptor, consistent with staining observed with CA1974 (Supplementary Fig. 1a). The cell line was con-firmed to be negative for mycoplasma.

**Size Exclusion-HPLC analyses.** Human TNF trimer (20 μM) in PBS pH 7.4 was pre-incubated with a 5-fold molar excess of **UCB-9260** in dimethyl sulfoxide (DMSO) (final concentration 2%) for 18 h at 4 °C. To this complex was added CA1974 mouse Fab (70 μM) in PBS pH 7.4 at a range of different molar excesses and maintained at room temperature for 2 h. As a control we also tested CA1974 binding to apo TNF which had been pre-incubated with DMSO only (final con-centration 2%). Samples were analysed by size exclusion-HPLC by injection of 25 μL onto a Superdex S200 10/300 (30 cm × 10 mm; 13–15 μm; GE Healthcare) column connected in series to an Agilent 1100 HPLC and eluted isocratically with PBS pH 7.4 at 1.0 ml/min for 30 min. Control samples containing Fab only and TNF-**UCB-9260** complex only were also prepared for calibration of the size exclusion-HPLC analysis. The molar excess at which unbound Fab appeared was equated to the saturation of binding.

**Crystal structure determination TNF.** The soluble form of human TNF (CID2043, UniProt P01375, residues 77-233) was generated as described above. The product contained an N-terminal serine residue as a cloning artefact but this was not part of the native sequence of the TNF molecule. The extracellular domain of human TNFR1 (CID5602, UniProt P19438, residues 42-184) with N54D and C182S mutations was expressed as a secreted protein in baculovirus-infected *Tri-choplusia ni* insect cells (Expression Systems, catalogue number 94-002 F and 91-002) using ESF-921 insect cell media (Expression Systems, catalogue number 96-001-01). An extra glycine was present at the N-terminus of the protein due to a cloning artifact. This is not part of the native sequence of the TNFR1. The fusion protein plasmid was cloned into the pEMB50 expression vector, which encodes a cleavable N-terminal secretion signal and His-tagged fusion protein. Virus was generated using the baculovirus expression system. Infected insect cells secreted the fusion protein into the media. The fusion protein was purified by Ni-NTA chelate chromatography and eluted from the Ni column using an imidazole gradient. The eluted protein was cleaved with protease to release the N-terminal His-fusion tag. The cleaved TNFR1 was subsequently purified by a subtractive Ni chelate chro-matography step and further purified by size exclusion chromatography. The final TNFR1 product was typically concentrated to 10.0 mg/ml and flash-frozen in liquid nitrogen.

To form the complex, 333 μl of purified human TNF at 300 mM was mixed with 4567 μl of SEC buffer (10 mM HEPES, pH 7.5, 150 mM NaCl) and 100 μL of **UCB-8733** (10 mM in DMSO, approximately 10 molar excess) and incubated at 4 °C overnight. The following day, the complex was formed by adding 4080 μl of SEC buffer to 700 μl TNFR1, 5,000 μl of TNF/**UCB-8733** mix, and 220 μl CA1974 Fab fragment at 500 mM. Total volume of the reaction was 10 ml with a final molar ratio of 3 TNF monomers (equivalent to 1 trimer): 2.5 TNFR1 receptors: 1.2 Fab. The ternary complex (cytokine, small molecule inhibitor, receptor, and Fab) was incubated for 1 h at 4 °C. Analytical SEC was used to assess complex formation (Supplementary Fig. 2). The sample was then concentrated to 1.5 ml and was loaded in a single injection on Superdex 200 16/600 size exclusion column (120 ml) pre-equilibrated with 10 mM HEPES pH 7.5, 150 mM NaCl. Peak fractions of the ternary complex were selected and concentrated to 13.7 mg/ml and immediately used in crystallization trials. The ternary complex was crystallized by sitting drop vapour diffusion by mixing 0.5 μl of complex with 0.5 μl of Wizard III/IV: 0.1 M HEPES, pH7.0, 10% PEG6,000 (condition B8) over 80 μl of the same crystallization solution. Crystals were harvested for data collection approximately 2 months after initial set-up. They were cryo-protected in glycerol performed in 5-10-15% steps, then frozen directly in liquid nitrogen for data collection performed at Advanced Photon Source at Argonne National Laboratory, Life Sciences Collaborative Access Team (LS-CAT), beamline 21-ID-F (wavelength 0.9786 Å, 100 K).

Crystallization of TNF with **UCB-8733** was achieved using methods previously described[5]. In brief, TNF at 4-7 mg/ml was incubated with 0.5 mM **UCB-8733** overnight at 4 °C. Crystals were grown at 16 °C by sitting drop vapour diffusion with a mixture of 0.5 μl protein:compound and 0.5 μl of 24.4% w/v PEG 3350, 0.1

M HEPES, pH 7.0. Crystals were cryo-protected in 10% ethylene glycol and frozen in liquid nitrogen for data collection. The data set was collected at APS, LS-CAT, beamline 21-ID-G (wavelength 0.97872 Å, 100 K). The structure of the human TNF (CID2043), human TNFR1 (CID5602), and CA1974 Fab complex with small-molecule **UCB-8733** was solved by molecular replacement using Phenix.Phaser[29] with input models based upon previously determined unpublished structures (Supplementary Table 3). Data were integrated in XDS and scaled using XSCALE[30]. Initial structure determination and refinement used data to 3.00 Å resolution from a single crystal. Iterative manual model building using Coot[31] and in Phenix.Refine[29] continued until $R$ and $R_{free}$ reached $R = 0.201$, $R_{free} = 0.259$ (Ramachandran favoured = 97.55%, outliers = 0.25%). Diffraction data for TNF and small-molecule **UCB-8733** were reduced and scaled as above and the structure was solved by rigid body refinement in Refmac[32] using a previously determined structure as the input model. Model building was performed using iterative rounds of Coot and Refmac until $R$ and $R_{free}$ converged at 0.184/ 0.270, respectively (Ramachandran favoured = 97.24%, outliers = 0.09%). Model quality was validated using Coot and MolProbity prior to deposition in the Protein Data Bank (PDB codes 7KPB/7KPA)[33–35]. Final data processing and refinement statistics are listed in Supplementary Table 3. Electron density images are shown in Supplementary Fig. 3 (human TNF [bound with **UCB-8733**] in complex with human TNFR1 and CA1974 Fab) and 6 (human TNF bound with **UCB-8733**).

**Assessment of CA1974 as a target occupancy reagent.** We initially tested the ability of CA1974 to detect a range of different recombinant human TNF con-centrations pre-incubated with saturating levels of small molecule. Pre-incubation of TNF with excess NCE was achieved by mixing 10 μg/ml TNF with either 1% (v/ v) DMSO or 10 μM **UCB-8733** in 0.1% (w/v) BSA/PBS, resulting in an approxi-mately 50-fold molar excess of small molecule inhibitor over TNF. Following overnight incubation the complex was diluted to a range of concentrations in human plasma that had been depleted of endogenous TNF by immunoprecipita-tion using an in-house anti-TNF monoclonal antibody. Small molecule-bound TNF was measured using a sandwich ELISA. A Nunc 96 well maxisorp ELISA plate (Thermo Fisher, catalogue number M9410) was coated with 0.1 μg/ml CA1974 in PBS overnight at 4 °C. After washing three times in wash buffer (0.05% [v/v] Tween-20/PBS) the plate was blocked in 2% (w/v) BSA/PBS for at least 1 h at RT. Each sample (100 μl) was loaded to the blocked ELISA plate and incubated for 30 min at room temperature (RT) with shaking at 200 rpm (Stuart Scientific Orbital Shaker SO3). The plate was washed three times in wash buffer before adding 100 μl per well of biotinylated anti-TNF detection antibody (R&D Systems, catalogue number BAF210), at a concentration of 0.1 μg/ml in 1% (w/v) BSA/wash buffer, for 1 h at RT with shaking at 200 rpm. Following three washes, 100 μl per well of 100 ng/ml Streptavidin-horseradish peroxidase (Sigma Aldrich, catalogue number RABHRP3-600UL) in 1% BSA/wash buffer was added and incubated for 45 min at RT with shaking at 200 rpm. The plate was washed three times in wash buffer before including a biotin-tyramide (B-T) amplification step to provide additional amplification, using the ELAST system from Perkin Elmer (catalogue number NEP116001EA). B-T solution was diluted 1 in 100 in ELAST buffer, 100 μl added per well and the plate incubated for exactly 15 min at RT with shaking at 200 rpm before washing four times in wash buffer. An incubation with ELAST SA-HRP was performed, followed by four washes and the addition of 100 μl TMB per well. When appropriate colour change had been reached, the reaction was stopped by the addition of 100 μl per well of 2.5 M sulphuric acid. Absorbance at 450 nm was measured using a Thermo Multiskan EX plate reader with Ascent Software version 2.6.

To determine the concentration at which TNF plus excess small molecule inhibitor could be detected above apo TNF background, three standard deviations of the TNF plus DMSO OD results were added to the mean of these values. All TNF plus small molecule spike results greater than this value were considered to be significantly different to apo TNF, with a probability of at least 0.999.

**Reporting summary.** Further information on research design is available in the Nature Research Reporting Summary linked to this article.

## Data availability
Relevant data sets generated and/or analysed during the current study are provided in a source data file. The structures reported in this paper have been deposited to the PDB under the accession codes 7KPA and 7KPB. Source data are provided with this paper.

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

## Acknowledgements

We would like to thank the following UCB Pharma personnel for their contributions to the research and manuscript: Catherine Simpson for help with flow cytometry and figure preparation, Grace Smith for help with DNA sequencing, Joanne Compson for Biacore support, Hanna Hailu for antibody expression, Geofrey Odede for antibody purification, Katarina Gore for assistance with statistical analyses and Martin Lowe for providing information regarding small molecule synthesis. We would also like to thank personnel at Beryllium Discovery Corp. for their contributions to the research and manuscript. Special contributions from Jan Abendroth whose work set the foundation for structures described; James W. Fairman, Tracy L. Arakaki, and Deborah G. Conrady for crystallography and structure determination support. We also thank Sanofi for the target occupancy discussions and for their financial support.

## Author contributions

R.M., B.C., A.T., A.S.-T., E.S.H., A.S., and D.F. III produced, analysed, and interpreted data for the paper. A.M. provided guidance on mode of action experiments, helped produce figures for the paper, and had input on writing. T.C. and D.M. analysed and interpreted crystallography data and helped produce structural figures. D.J.L managed the antibody discovery work. D.J.L. designed and wrote the article with input primarily from D.M. and A.D.G.L. The idea of using antibodies as target-engagement biomarkers was originally conceived by J.P., J.O'C., and A.D.G.L. The TNF small molecule project was managed by T.B. and J.O'C., both provided input into the paper writing.

## Competing interests

D.J.L., J.P., D.M., B.C., A.T., A.M., T.C., T.B., J.O'C., and A.D.G.L. are/were all employees of UCB Pharma and may hold stock and/or stock options. A.S-T., E.S.H., and A.S. are/were employees of UCB Pharma.
