## [Peer Review File · Nature Communications]

Reviewers' comments:

Reviewer #1 (Remarks to the Author):

Lightwood et al. present a study describing the generation and characterization of a monoclonal antibody that was raised to a small molecule bound form of hTNF- α . Apo-TNF- α is a symmetric trimeric molecule. The two small molecules described in the manuscript are among a number that induce a global symmetric to asymmetric conformational transition. This perturbed structure forces a breakdown of the functional hormone-receptor stoichiometry leading to inhibition of the signaling capacity of the system. The antibody (CA1974) shows a binding discrimination between the apo- vs. the small molecule-loaded conformations of about 2-3 orders of magnitude. The authors purport that with this level of selectivity coupled with its high affinity, it should be possible to quantify the amount of compound-loaded TNF- α species existing within a mixture of conformational states in complex biological samples.

This paper is one of a trilogy of papers apparently submitted simultaneously by some combination of members of an industrial consortium. All papers describe some facet of the discovery and characterization of a cohort of small molecule moieties that induce the symmetric to asymmetric transition of TNF- α and its resulting biological implications. While carefully reading the other two preprints, this reviewer did not dig into all their experimental details. However, suffice it to say, they are highly connected in theme and the paper under review here is to an extent, an extension of the other two. Its uniqueness is in the potential applications of the antibody.

The CA1974 Fab fragment was used as a crystallization chaperone to crystallize the complex containing the hTNF- α trimer ligated with UCB-8733 and the hTNFR1 receptor. This Fab- complex was purported to represent the first reported crystal structure of hTNF- α with hTNFR1 since all previous attempts to crystallize it had failed. This demonstrated the usefulness of the Fab as a crystallization chaperone, but the structure of a similar small molecule structure of murine (m)TNF- α with hTNFR1 was reported in Paper #2 (McMillan et al.). No comparison was made between that complex and the one reported in this manuscript. McMillan et al. make a compelling argument that the murine and human TNF- α structures when complexed to an asymmetrically inducing small molecule are virtually identical. So, the Fab structure with hTNF- α and hTNFR1 is really more of a demonstration of the usefulness of the Fab as a chaperone to crystallize a recalcitrant system, the structure itself does not provide any additional insights into the biology of the system.

Use of the Fab as a reporter of the small molecule loaded form of TNF- α in complex biological mixtures is an interesting application. It is well established that Fabs can stabilize transient conformational intermediates, as is the case for CA1974 reported here. In Paper #1 (O'Connell et al.), a molecular dynamics simulation suggested that the apo-TNF- α trimer exists in two most populated states (symmetric/asymmetric) separated by a 5 kcal/mol barrier. This is consistent with the SPR data showing that CA1974, which was generated to the asymmetrical conformation, binds better to that state by it at least 2 to 3 logs than the symmetric conformation. The difference in energy is presumably because binding this form requires the antibody to expend energy to

overcome this barrier. In fact, the differences in binding affinities are probably a good readout of the energy differences of these two populations in the apo-state. However, using the antibody as a tool to quantify the population of the small molecule bound form of TNF- α in biological mixtures is not straightforward. Conformationally selective antibodies are not “agnostic” in that they drive the equilibrium toward the bound state. Therefore, any readout would overestimate the population of the bound state in the mixture and would require a correction to compensate for this effect taking into consideration the amount of free compound there is in the mixture and its affinity (which is not given). This is seen in Fig. 1a where the curves between UBC-9260 and UCB-8733 clearly differ. Since both compounds presumably induce identical asymmetric conformations, the only difference can be in their relative affinities. If some other effect is in play, then that should be explained. It is also a bit surprising that some binding was not observed to the apo-complex because it binds in the single nM range itself. Supplementary Fig. 3 is convincing in showing small amounts of bound compound can be detected by the antibody and while it is a nice proof of concept, to adequately evaluate the feasibility as a real research tool, it would be necessary to challenge it with a biologically relevant type of complex mixture.

Some comments that could be considered are mentioned below:

- 1). A highly confusing element that runs through the paper is trying to keep track of the compounds used in the different experiments. In the results section, the reader is introduced to the compounds: UBC-9260 and UCB-8733. However, no affinities are provided. From Paper #2, if you dig deep enough, you find that UBC-9260 is an affinity matured form of UBC-5307. To find the affinity of UBC-9260, one has to go back to the other paper. I was not able to track down the origin of UCB-8733 or its affinity.
- 2). Fig. 3 (a) and (b) provide useful information. (c) and (d) do not. You just see a mass of stuff. What is the insight they provide? Similar issues with Fig. 5. (b) and (c) basically provide most of the pertinent information.
- 3). Apparently, the initial structure of the Fab complex was determined from a 3.0Å data set and then extended to 2.4Å using a second data set. Why bother with the lower resolution data set to begin with? What is pretty striking is that the final high resolution data set is highly over fit with the difference between R(work) and R(free) close to 9%. This is a red flag. So, the final R(work) of 18.4 is highly suspect. The lower resolution data set structure fits current accepted standards much better.

Reviewer #2 (Remarks to the Author):

The study of Lightwood et al is based on two other submitted manuscripts currently under review in Nature and Nature Structure Molecular Biology. In these two other manuscripts a group of small molecule inhibitors of TNF and their mode of action have been identified. Crystal structures of the inhibitor complexed with human TNF or murine TNF and the human TNFR1ed revealed that these novel inhibitors act by stabilizing an open distorted conformation of the TNF trimer which only interacts with two instead of three TNFR1 monomers thereby preventing the assembly of oligomeric TNF-TNFR1 signaling platforms.

Against this background, Lightwood et al isolated an antibody (CA1974) with high specificity for the inhibitor-TNF complex and very limited recognition of "free" TNF. The CA1974 Fab fragment not only allowed resolution of the TNF-TNFR1 structure by forming Fab-inhibitor-TNF-TNFR1ed crystals but can also be used to determine the concentration of inhibitor occupied TNF in biological samples.

In general, the triad of manuscripts describes straightforwardly the identification and mode of action of a novel class of small molecules inhibiting TNF-induced TNFR1 activation. Unfortunately, however, the basic molecular characterization of these novel TNF inhibitors is incomplete:

TNF exists in form of soluble trimers (investigated in the paper triad) but also as a transmembrane molecule (not covered in the manuscripts). Both forms of TNF furthermore do not only act via TNFR1 (investigated in the manuscript triad) but also via TNFR2 (not covered in the manuscripts). I think when novel "small molecules that inhibit TNF signaling" are described in a high impact journal, it is mandatory to present a comprehensive basic in vitro characterization of the effects of these inhibitors on TNF signaling. In this respect, I miss data showing i) the effect of the inhibitors on TNFR2 signaling and ii) the effect of the inhibitor on the capacity of transmembrane TNF to stimulate TNFR1 and TNFR2 signaling.

The data presented by the two other manuscripts appears to me of broad and general interest. Against the background of these two manuscripts, however, the present study is of more specialized nature. The CA1974 "chaperoned" inhibitor-hTNF-hTNFR1ed structure allows no major insights that go beyond those derived from the inhibitor-mTNF-hTNFR1ed complex presented in ref.2. CA1974 is certainly a fascinating antibody and could be of considerable relevance in the case of preclinical/clinical development of these novel inhibitors. For a broad readership, however, this is presumably less exciting.

Minor comments:

Figure 1: Data with a control antibody recognizing inhibitor-TNF complexes as well as free TNF would further strengthen the specificity data of CA1974.

Figure 2: mFab and hTNF-compound complex cannot be unambiguously distinguished in the SE-HPLC. I admit that it is plausible that the 15.5 ml peak, which appears with increasing Fab amounts,

is free Fab and not hTNF-compound complex but this could easily be controlled experimentally in a definite manner by WB analysis of the fractions.

Line 27: "to obtain" is perhaps better than "to produce"

Line 55: "TNF bound to human TNFR1" instead of "TNF bound to its cognate human TNFR1"

Line 64: I assume "sampled transiently by CA1974" is meant, not "by TNF α trimer"

Line 68: "into TNF α structure" appears more correct to me than "into the signaling of TNF α "

Line 235: "three receptor monomers" instead of "three receptor dimers"

There is frequently no blank space between measure and unit.

Reviewer #3 (Remarks to the Author):

The manuscript by Lightwood et al. described the development, characterization, and utility of a rat monoclonal antibody (mAb) that recognizes a unique epitope on an asymmetric TNF- α trimer induced by a small molecule inhibitor. The development of this small molecule inhibitor is reported in a separate manuscript (reference 1) and its mechanism of action pertaining to interference with TNF- α receptor-mediated signaling is reported in another manuscript (reference 2), all of which are currently under review. Collectively, the three manuscripts make an exciting and robust advance toward small molecule inhibitors of TNF- α with the potential to compete with blockbuster biologics, such as Humira and Enbrel, targeting the same pathway. While keeping the three manuscripts separate is clearly justified with minimal overlap, they provide the cornerstones of a compelling story backed up by sound biochemical, cell biology, and structural biology data. The mAb developed by Lightwood et al. using immunization with the TNF- α /small molecule inhibitor complex and negative screening against the apo-protein, afforded key proof that the drug induces an asymmetric epitope in the TNF- α trimer that is bound by the mAb. The authors used X-ray crystallography to solve the structure of this mAb (in Fab format) bound to the asymmetric TNF- α /small molecule inhibitor and two TNF- α receptor molecules that engage the unperturbed TNF- α monomers in the trimer. The mAb further allowed to elucidate the mechanism of action of the small molecule inhibitor by identifying a fraction of apo-protein that breathes through the asymmetric configuration, suggesting that this state is stabilized by the drug. Overall this is a highly relevant and well done and written study. A few minor comments: (1) Mention the resolution of the crystal structure in the results section; (2) in the materials and methods section, clarify that the recombinant mAb formats used in the study are chimeric rat/mouse IgG1 and Fab, i.e. are composed of rat variable domains and mouse constant domains; (3) in Fig. 1B the color choice is confusing as it

does not correspond to Fig. 1A; (4) in Suppl. Fig. 3, one might expect a higher background in the sandwich ELISA given the substantial fraction of breathing apoprotein - please explain the low background; did endogenous TNF-alpha have to be depleted due to breathing apoprotein background?

Responses to reviewers' comments:

Reviewer #1 (Remarks to the Author):

Lightwood et al. present a study describing the generation and characterization of a monoclonal antibody that was raised to a small molecule bound form of hTNF- α . Apo-TNF- α is a symmetric trimeric molecule. The two small molecules described in the manuscript are among a number that induce a global symmetric to asymmetric conformational transition. This perturbed structure forces a breakdown of the functional hormone-receptor stoichiometry leading to inhibition of the signaling capacity of the system. The antibody (CA1974) shows a binding discrimination between the apo- vs. the small molecule-loaded conformations of about 2-3 orders of magnitude. The authors purport that with this level of selectivity coupled with its high affinity, it should be possible to quantify the amount of compound-loaded TNF- α species existing within a mixture of conformational states in complex biological samples.

This paper is one of a trilogy of papers apparently submitted simultaneously by some combination of members of an industrial consortium. All papers describe some facet of the discovery and characterization of a cohort of small molecule moieties that induce the symmetric to asymmetric transition of TNF- α and its resulting biological implications. While carefully reading the other two preprints, this reviewer did not dig into all their experimental details. However, suffice it to say, they are highly connected in theme and the paper under review here is to an extent, an extension of the other two. Its uniqueness is in the potential applications of the antibody.

Re reviewer 1's comment re the generation of a "*trilogy of papers apparently submitted simultaneously by some combination of members of an industrial consortium*", we would like to clarify that the work described across these papers was carried out over several years by scientists at UCB Pharma. Some data were provided by Beryllium Discovery (a collaborator at the time of work but which is now a UCB Pharma company). We are in the process of submitting a number of papers that describe the various aspects of this ground-breaking project. The first two papers, O'Connell et al. and McMillan *et al.* are currently under review. The paper presented here on a conformation selective antibody, is the third paper in the series. There will be further communications in the future, one of which will detail the cellular mode of action of the small molecules and cover effects on membrane-bound TNF α and signalling from TNFR1 and TNFR2.

Regarding the following comment: *The CA1974 Fab fragment was used as a crystallization chaperone to crystallize the complex containing the hTNF- α trimer ligated with UCB-8733 and the hTNFR1 receptor. This Fab- complex was purported to represent the first reported crystal structure of hTNF- α with hTNFR1 since all previous attempts to crystallize it had failed. This demonstrated the usefulness of the Fab as a crystallization chaperone, but the structure of a similar small molecule structure of murine (m)TNF- α with hTNFR1 was reported in Paper #2 (McMillan et al.). No comparison was made between that complex and the one reported in this manuscript. McMillan et al. make a compelling argument that the murine and human TNF- α structures when complexed to an asymmetrically inducing small molecule are virtually identical. So, the Fab structure with hTNF- α and hTNFR1 is really more of a demonstration of the usefulness of the Fab as a chaperone to crystallize a*

recalcitrant system, the structure itself does not provide any additional insights into the biology of the system. Although the structures of the complexes described in McMillan and Lightwood *et al.* are similar, we believe that the structure presented in the current paper unambiguously confirms the nature of the complex consisting of the human cytokine and human receptor subunits. Although it was reasonable to assume that the mouse TNF α –human receptor system could be a surrogate, it is still noteworthy to confirm in the current paper the structure of the human TNF-human receptor complex, which is a world first and at the same time demonstrate the usefulness of this antibody in supporting crystallisation experiments. We agree with the reviewer’s comment that the paper highlights the utility of the antibody as a chaperone in the crystallisation of a recalcitrant system. We hope that this aspect of the antibody’s value as a tool reagent, in addition to its use as a detection agent to help determine mode of action of compounds, to measure target occupancy and to provide insight into the molecular dynamics of TNF α is clear in the paper. Consistent with the reviewer’s comments, the chaperone component is valuable, but in the context of this paper may be considered of less importance than the other aspects of this remarkable antibody. We have modified the manuscript abstract, introduction, results and discussion to reflect this desired focus. We have also included the “target occupancy data” in the main figures (now Fig. 6, originally supplementary Fig. 4). To clarify the similar structure between the one reported here and that produced by McMillan *et al.*, we have included an overlay in the supplementary section (supplementary Fig. 3).

Re this comment: *Use of the Fab as a reporter of the small molecule loaded form of TNF- α in complex biological mixtures is an interesting application. It is well established that Fabs can stabilize transient conformational intermediates, as is the case for CA1974 reported here. In Paper #1 (O’Connell *et al.*), a molecular dynamics simulation suggested that the apo-TNF- α trimer exists in two most populated states (symmetric/asymmetric) separated by a 5 kcal/mol barrier. This is consistent with the SPR data showing that CA1974, which was generated to the asymmetrical conformation, binds better to that state by it at least 2 to 3 logs than the symmetric conformation. The difference in energy is presumably because binding this form requires the antibody to expend energy to overcome this barrier. In fact, the differences in binding affinities are probably a good readout of the energy differences of these two populations in the apo-state. However, using the antibody as a tool to quantify the population of the small molecule bound form of TNF- α in biological mixtures is not straightforward. We agree with the comments made by the reviewer regarding the energy barriers etc. and we also acknowledge the reviewer’s recognition of the utility of the antibody as a “biomarker” and we would like to draw their attention to Fig. 6 (in the new version of the manuscript) which clearly shows that CA1974 is able to measure TNF-small molecule complex in human plasma. No reactivity to apo TNF α was observed. This also addresses a subsequent comment (i.e. Supplementary Fig. 3 is convincing in showing small amounts of bound compound can be detected by the antibody and while it is a nice proof of concept, to adequately evaluate the feasibility as a real research tool, it would be necessary to challenge it with a biologically relevant type of complex mixture).*

Re this comment: *Conformationally selective antibodies are not “agnostic” in that they drive the equilibrium toward the bound state. Therefore, any readout would overestimate the population of the bound state in the mixture and would require a correction to compensate for this effect taking into consideration the amount of free compound there is in the mixture and its affinity (which is not given). This is seen in Fig. 1a where the curves between UBC-9260 and UCB-8733 clearly differ. Since*

both compounds presumably induce identical asymmetric conformations, the only difference can be in their relative affinities. If some other effect is in play, then that should be explained. It is also a bit surprising that some binding was not observed to the apo-complex because it binds in the single nM range itself. In principle, we agree with the comments of the reviewer, however, it should be noted that only a very small fraction of the apo TNF α is in the asymmetric state (based on DEER experiments, 6% of trimers are sampling a single “open” interface at any one time (Carrington et al., 2017 - [http://www.cell.com/biophysj/pdf/S0006-3495\(17\)30628-8.pdf](http://www.cell.com/biophysj/pdf/S0006-3495(17)30628-8.pdf))) and that the affinity of CA1974 for apo TNF α form is low compared to the complex (2-3 orders of magnitude lower). In addition the association rate of small molecule into apo TNF α is also very slow (UCB-9260 = $3.1E+03 \text{ M}^{-1}\text{s}^{-1}$; UCB8733 = $4.5E+03 \text{ M}^{-1}\text{s}^{-1}$). Taken together, we would suggest that during the relatively short assay capture step (in this example 30 minutes is used) for the reasons outlined above, there would be minimal impact of “in assay association” on the measurement of bound NCE-TNF α in the sample. Reducing the duration of the capture step would further minimise this potential effect.

It should be noted that details of the clinical assay with further data will be reported in a separate future publication. The observations highlighted by the reviewer are duly noted and we will endeavour to establish and control for the impact of this “in-assay association” effect as part of the clinical assay work up. We would however like to highlight that as part of a clinical assay using CA1974, a standard curve would always be included comprising recombinant TNF α complexed with the relevant compound. All plates would contain a standard curve, such that all samples and standards would be treated in an identical manner. The standard curve would be optimised to ensure that Apo- TNF α is not detected within the concentration range of the assay, and an Apo- TNF α control would be included in every assay to demonstrate that it is not detected. A measurement of total TNF α (i.e. bound + Apo) would also be important for a clinical assay, such that bound- TNF α levels could be normalised in individual patient samples. As indicated, details of this will be included in the aforementioned later clinical assay paper.

We would like to clarify that the current paper, purely describes the potential utility of this remarkable antibody to act as a target engagement biomarker to measure TNF-NCE complex in complex biological samples rather than providing details of a clinical assay.

Regarding the comment about it being surprising that no binding is observed in the ELISA screens (Fig. 1 and Fig. 6 (in the new version of the manuscript)), as indicated above, the low percentage of asymmetric trimer present in the sample, in addition to the low affinity, likely accounts for the lack of ELISA signal with apo TNF. To clarify the observation made by the reviewer, we will add a sentence to the discussion summarising the above explanation. We thank the reviewer for highlighting this.

To address the comment “where the curves between UCB-9260 and UCB-8733 clearly differ”, We thank the reviewer for drawing our attention to this. We have since reviewed the data and expanded the dataset to now include a further experiment (n=2, each performed in duplicate). This has produced a new curve in fig 1 (a) which demonstrates that the binding of CA1974 Fab to both TNF-9260 and TNF-8733 are comparable.

To help keep track of the two compounds used in this study, we have added a table to the supplementary section which summarises the structure and affinity of UCB-9260 and UCB-8733 (supplementary figure 7 in revised manuscript). We thank the reviewer for helping us enhance the clarity of the manuscript.

Some comments that could be considered are mentioned below:

1). A highly confusing element that runs through the paper is trying to keep track of the compounds used in the different experiments. In the results section, the reader is introduced to the compounds: UBC-9260 and UBC-8733. However, no affinities are provided. From Paper #2, if you dig deep enough, you find that UBC-9260 is an affinity matured form of UBC-5307. To find the affinity of UBC-9260, one has to go back to the other paper. I was not able to track down the origin of UBC-8733 or its affinity. As indicated above, to help keep track of the two compounds used in this study, we have added a table to the supplementary section which summarises the structure and affinity of UCB-9260 and UCB-8733 (supplementary figure 7 in revised manuscript). We thank the reviewer for helping us enhance the clarity of the manuscript.

2). Fig. 3 (a) and (b) provide useful information. (c) and (d) do not. You just see a mass of stuff. What is the insight they provide? Similar issues with Fig. 5. (b) and (c) basically provide most of the pertinent information. It is not completely clear what the reviewer is asking regarding Fig. 3 given there is only (a), (b) and (c), and not (d). However, we have clarified the figure legend to highlight what is shown in Fig. 3(c). i.e. that the zoomed-in structure shows side chain clashes between the CA1974 Fab CDR regions and the symmetric apo TNF. The description of this is given in page 7 of the revised manuscript and we have modified the text to clarify that the view in Fig.3(c) is with symmetric apo TNF α . Further detail is also provided in supplementary Fig. 2. Re Fig 5, we feel that the figure clearly shows the epitope of CA1974 on human TNF α (Fig. 5 (a) and (c)). Fig. 5 (c) and (d) show the amino acids which vary in mouse (c) and cynomolgus (d) at this epitope. The exact amino acid differences are shown in supplementary table 2. The Fig. 5 legend has now been modified slightly to clarify this and to reference supplementary table 2. As explained in the text, these differences likely account for the different observed affinities between species, particularly between human and mouse TNF α . We thank the reviewer for giving us the opportunity to clarify the manuscript.

3). Apparently, the initial structure of the Fab complex was determined from a 3.0Å data set and then extended to 2.4Å using a second data set. Why bother with the lower resolution data set to begin with? What is pretty striking is that the final high resolution data set is highly over fit with the difference between R(work) and R(free) close to 9%. This is a red flag. So, the final R(work) of 18.4 is highly suspect. The lower resolution data set structure fits current accepted standards much better. Thank you for carefully reviewing the structural components of this paper. The initial dataset collected was TNF bound to UCB-8733 at 2.4Å, which was later used as a guide for the complex structure of TNF, Fab, Receptor and the same compound, determined at 3.0Å. The original structure at 2.4Å was refined several years ahead of publication (2012 vs 2016). We have reordered the statistics in the table (supplementary table 3) to reflect the order which they were collected.

We also revisited this dataset to identify any issues that may be leading to a higher than normal difference between R and R_f (as you referenced, 9% was observed). Dataset statistics suggested that the last 5 images had significantly higher R_{meas} per frame than the first 135 images collected. Removing these last 5 images, along with increasing the resolution cutoff from 2.4 to 2.3Å based on I/SigI >2.5, R-free fraction from 5 to 10%, followed by MR and refinement in Phenix.refine, and reduction of waters from ~150 to 63, resulted in an R/R_{free} <5%, without resulting in significant

differences to the model.

Please also note that validation reports for both the antibody-TNF-UCB-8733-receptor complex structure and the TNF-UCB-8733 structure have been included in the resubmission.

Reviewer #2 (Remarks to the Author):

The study of Lightwood et al is based on two other submitted manuscripts currently under review in Nature and Nature Structure Molecular Biology. In these two other manuscripts a group of small molecule inhibitors of TNF and their mode of action have been identified. Crystal structures of the inhibitor complexed with human TNF or murine TNF and the human TNFR1ed revealed that these novel inhibitors act by stabilizing an open distorted conformation of the TNF trimer which only interacts with two instead of three TNFR1 monomers thereby preventing the assembly of oligomeric TNF-TNFR1 signaling platforms.

Against this background, Lightwood et al isolated an antibody (CA1974) with high specificity for the inhibitor-TNF complex and very limited recognition of “free” TNF. The CA1974 Fab fragment not only allowed resolution of the TNF-TNFR1 structure by forming Fab-inhibitor-TNF-TNFR1ed crystals but can also be used to determine the concentration of inhibitor occupied TNF in biological samples.

In general, the triad of manuscripts describes straightforwardly the identification and mode of action of a novel class of small molecules inhibiting TNF-induced TNFR1 activation. Unfortunately, however, the basic molecular characterization of these novel TNF inhibitors is incomplete:

TNF exists in form of soluble trimers (investigated in the paper triad) but also as a transmembrane molecule (not covered in the manuscripts). Both forms of TNF furthermore do not only act via TNFR1 (investigated in the manuscript triad) but also via TNFR2 (not covered in the manuscripts). I think when novel “small molecules that inhibit TNF signaling “ are described in a high impact journal, it is mandatory to present a comprehensive basic in vitro characterization of the effects of these inhibitors on TNF signaling. In this respect, I miss data showing i) the effect of the inhibitors on TNFR2 signaling and ii) the effect of the inhibitor on the capacity of transmembrane TNF to stimulate TNFR1 and TNFR2 signaling. We acknowledge the reviewer’s comments and as highlighted, confirm that the first two papers (O’Connell et al. and McMillan et al.) and the current paper, focus on the description of small molecule inhibitors of soluble TNF. This is clearly stated in the abstract of O’Connell et al. – *“Here we report the discovery of potent small molecule inhibitors of TNF α that stabilise an asymmetrical form of the soluble TNF trimer.”* In order to clarify the focus of the current paper on soluble TNF, we have amended the abstract to accurately reflect this. The first two papers are still undergoing the review process, but to date the focus on soluble TNF α has been accepted. We do however completely agree with the reviewer’s comments about the important role of membrane- TNF α and TNFR2. This aspect of TNF α biology and the mode of action of the small molecule inhibitors has been an intense area of study for the research team at UCB. As such, given the large body of data that has been generated, a further paper covering the cellular mode of action work is currently in preparation. This will represent a fourth paper in the series. This further mode of action paper confirms the inhibitory effect of small molecules on soluble TNF α signalling through TNFR1 but also reports on the effect of the small molecules on membrane TNF and signalling

through TNFR2. In this cellular mode of action paper, the CA1974 antibody is used as a reagent to help with this characterisation. We hope that the reviewer will accept this publication strategy and recognise that it is always challenging to decide how to separate high value data to produce informative and interesting publications. We have rigorously discussed this publication strategy and believe we have packaged the data in the most optimum way to tell this compelling story.

The data presented by the two other manuscripts appears to me of broad and general interest. Against the background of these two manuscripts, however, the present study is of more specialized nature. The CA1974 “chaperoned” inhibitor-hTNF-hTNFR1ed structure allows no major insights that go beyond those derived from the inhibitor-mTNF-hTNFR1ed complex presented in ref.2. CA1974 is certainly a fascinating antibody and could be of considerable relevance in the case of preclinical/clinical development of these novel inhibitors. For a broad readership, however, this is presumably less exciting. We do agree with the reviewer that CA1974 is a fascinating antibody and as indicated in the introduction to this “reviewer response” letter, we feel that this paper highlights how CA1974 has supported the discovery of a small molecule inhibitor against TNF, one of the most well validated and studied therapeutic targets in the autoimmune and inflammatory space. The antibody has helped to define the mode of action and specificity of the small molecule inhibitors allowing us to demonstrate that receptor binding on cells is not completely abolished. In addition, this remarkable antibody has facilitated the measurement of target occupancy in complex biological samples, revealed the molecular dynamics of the apo TNF α trimer and its ability to transiently adopt this open conformation and acted as a chaperone in crystallisation studies. We believe this work defines a new role for monoclonal antibodies as tools to facilitate the discovery and development of small molecule inhibitors of protein-protein interactions (PPIs), a potentially transformational emerging class of drug molecules. As such, we feel that this paper will have broad interest to the field of biology and chemistry particularly those interested in drug discovery and development, protein dynamics, structural biology and antibody biology. We have refocused the paper to clarify the utility of this antibody in these regards. We hope the reviewer and the journal agree that this will represent a landmark paper highlighting the synergy between small molecule and antibody research in PPI drug discovery, a field which is viewed as “high-hanging fruit”, a theme originally suggested in a forward looking Perspective in 2012 (Lawson, Nature Reviews Drug Discovery 11, 519-525).

Minor comments:

Figure 1: Data with a control antibody recognizing inhibitor-TNF complexes as well as free TNF would further strengthen the specificity data of CA1974. We do have this data but decided not to include it due to the amount of data that is already in the manuscript. For the information of the reviewer we have included a figure at the bottom of this letter which shows the use of a TNF-specific polyclonal antibody (Invitrogen (AHC3812)) as a capture agent for the measurement of both apo TNF α and TNF α complexed separately with three small molecule inhibitors that are related to the compounds described in this paper. The protocol was the same as that used to generate the data in Fig. 6 (based on figure number in new manuscript) (i.e. spiked into human plasma). As indicated, this antibody binds similarly to both the apo and the small molecule-bound form of TNF. This antibody, or a similar capture agent, will ultimately be used to measure “total TNF” in clinical assays. Such data will likely be included in a subsequent clinical assays paper (see above comment to Reviewer 1). In addition we also have data (n=1) showing that Adalimumab (a commercial anti- TNF α therapeutic antibody)

binds similarly to both apo TNF α and TNF α -small molecule complex captured via TNFR1 in an ELISA (i.e. same protocol as is used to generate the data shown in figure 1). These data are also included at the bottom of the letter.

Figure 2: mFab and hTNF-compound complex cannot be unambiguously distinguished in the SE-HPLC. I admit that it is plausible that the 15.5 ml peak, which appears with increasing Fab amounts, is free Fab and not hTNF-compound complex but this could easily be controlled experimentally in a definite manner by WB analysis of the fractions. We do not have the WB data to support this observation. We will modify the language in the paper to read.... *“As can be seen in Fig. 2, at equimolar concentrations of Fab and TNF α -small molecule complex, a new single higher molecular weight peak corresponding to Fab bound to TNF α trimer-compound complex was evident (retention time of approx. 13.5 mins) (representing 90.3% total protein). At 1.5x and 2x molar excesses of Fab, there was no change to the size of the complex (as judged by retention time) and there was a respective increase in the area of the peak present at approx. 15.5 mins likely representing free Fab.”* We thank the reviewer for bringing this aspect to our attention and we are pleased that we had the opportunity to clarify this section.

Line 27: “to obtain” is perhaps better than “to produce”

We have changed the text in the revised manuscript accordingly. We thank the reviewer for this suggestion.

Line 55: “TNF bound to human TNFR1” instead of “TNF bound to its cognate human TNFR1”

We have changed the text in the revised manuscript accordingly. We thank the reviewer for this suggestion.

Line 64: I assume “sampled transiently by CA1974” is meant, not “by TNF α trimer”

We are referring to the apo TNF α trimer naturally adopting this open state transiently rather than CA1974 binding this form. We have modified the language in this sentence to clarify this and used the word “adopted” rather than “sampled”: *“This supports the possibility that the perturbed open conformation exists naturally and is adopted transiently by the apo TNF α trimer and is consistent with previous molecular dynamic....”*. We thank the reviewer for highlighting this potentially confusing terminology and we now feel the description in the paper is improved.

Line 68: “into TNF α structure” appears more correct to me than “into the signaling of TNF α ”

We have modified the text to read: *“In addition to providing invaluable insights into the structure and dynamics of TNF α and the mode of action of the small molecules, the antibody....”*. We thank the reviewer for providing us with the opportunity to clarify the manuscript.

Line 235: “three receptor monomers” instead of “three receptor dimers”

We have changed the text to read: “*Consistent with previous observations that small molecules do not ablate engagement with all three receptor subunits...*”. We hope this change helps to clarify the manuscript.

There is frequently no blank space between measure and unit.

We have corrected this in the revised manuscript. We thank the reviewer for bringing this to our attention.

Reviewer #3 (Remarks to the Author):

The manuscript by Lightwood et al. described the development, characterization, and utility of a rat monoclonal antibody (mAb) that recognizes a unique epitope on an asymmetric TNF-alpha trimer induced by a small molecule inhibitor. The development of this small molecule inhibitor is reported in a separate manuscript (reference 1) and its mechanism of action pertaining to interference with TNF-alpha receptor-mediated signaling is reported in another manuscript (reference 2), all of which are currently under review. Collectively, the three manuscripts make an exciting and robust advance toward small molecule inhibitors of TNF-alpha with the potential to compete with blockbuster biologics, such as Humira and Enbrel, targeting the same pathway. While keeping the three manuscripts separate is clearly justified with minimal overlap, they provide the cornerstones of a compelling story backed up by sound biochemical, cell biology, and structural biology data. We gratefully acknowledge the kind comments of the reviewer and the recognised impact that this work could bring to the field, and potentially to patients suffering with serious disease.

The mAb developed by Lightwood et al. using immunization with the TNF-alpha/small molecule inhibitor complex and negative screening against the apoprotein, afforded key proof that the drug induces an asymmetric epitope in the TNF-alpha trimer that is bound by the mAb. The authors used X-ray crystallography to solve the structure of this mAb (in Fab format) bound to the asymmetric TNF-alpha/small molecule inhibitor and two TNF-alpha receptor molecules that engage the unperturbed TNF-alpha monomers in the trimer. The mAb further allowed to elucidate the mechanism of action of the small molecule inhibitor by identifying a fraction of apoprotein that breathes through the asymmetric configuration, suggesting that this state is stabilized by the drug. Overall this is a highly relevant and well done and written study. Again, we appreciate the reviewer's feedback and recognition of this exciting work.

A few minor comments:

(1) Mention the resolution of the crystal structure in the results section;

We have amended the results section accordingly. i.e. *“All components of the complex were well resolved at 3.0 Å, clearly indicating a single copy of CA1974 bound at the distorted interface of the TNF α trimer and copies of hTNFR1 bound at the non-distorted receptor binding sites (Fig. 3a)”*. We thank the reviewer for highlighting this omission.

(2) in the materials and methods section, clarify that the recombinant mAb formats used in the study are chimeric rat/mouse IgG1 and Fab, i.e. are composed of rat variable domains and mouse constant domains;

We have modified the text in the material and methods section to reflect this. We thank the reviewer for highlighting this and allowing us to clarify the manuscript.

(3) in Fig. 1B the color choice is confusing as it does not correspond to Fig. 1A;

We agree to the extent that the inclusion of the key-legend that appears to the right of the curves in Fig 1(a) provides some confusion given the same colour scheme is used for both 1(a) and 1(b). For this reason we have removed this key-legend. This will require the reader to refer to the description of the figures in the main legend. We have decided not to change the colour scheme in Fig 1(b) as we feel that red, green and blue work well and provide good clarity but hopefully removal of the key-legend removes some of the confusion.

(4) in Suppl. Fig. 3, one might expect a higher background in the sandwich ELISA given the substantial fraction of breathing apoprotein - please explain the low background; did endogenous TNF-alpha have to be depleted due to breathing apoprotein background?

This aspect has been summarised above in the response to reviewer 1's similar comment. Briefly, the low percentage of asymmetric trimer present in the sample (DEER experiments with spin-labelled TNF α which predicts 6% of trimers at any one time adopt a naturally-sampled asymmetric open state - Carrington et al., 2017) in addition to the low affinity (2-3 orders of magnitude lower than the TNF-small molecule complex), and the relatively short duration of the assay (30 mins) likely accounts for the lack of ELISA signal with apo TNF. To clarify the ELISA observation, we will add a sentence to the discussion summarising the above explanation. We thank the reviewer for again highlighting this aspect.

Supporting figures

Figure showing the use of a TNF-specific polyclonal antibody (Invitrogen #AHC3812) for the measurement of apo TNF α (blue) and TNF α complexed separately with three small molecules (red, green and purple)

Figure showing the binding of Adalimumab to apo TNF α (red) and TNF α + UCB-9260 complex (green) using a TNFR1-capture ELISA. (n=1)

Reviewers' comments:

Reviewer #1 (Remarks to the Author):

The changes made in the revision are adequate.